# Downhole distributed acoustic seismic profiling at Skytrain Ice Rise, West Antarctica

Alex M. Brisbourne[1], Michael Kendall[2], Sofia-Katerina Kufner[1], Thomas S. Hudson[2] and Andrew M. Smith[1],

[1]IDP, NERC British Antarctic Survey, Cambridge, CB3 0ET, UK
[2]Department of Earth Sciences, University of Oxford, Oxford, OX1 3AN, UK

*Correspondence to*: Alex M. Brisbourne (aleisb@bas.ac.uk)

**Abstract.** Antarctic ice sheet history is imprinted in the structure and fabric of the ice column. At ice rises, the signature of ice flow history is preserved due to the low strain rates inherent at these independent ice flow centres. We present results from a distributed acoustic sensing (DAS) experiment at Skytrain Ice Rise in the Weddell Sea Sector of West Antarctica, aimed at delineating the englacial fabric to improve our understanding of ice sheet history in the region. This pilot experiment demonstrates the feasibility of an innovative technique to delineate ice rise structure. Both direct and reflected P- and S-wave energy, as well as surface wave energy, are observed using a range of source offsets, i.e., a walkaway vertical seismic profile, recorded using fibre optic cable. Significant noise, which results from the cable hanging untethered in the borehole, is modelled and suppressed at the processing stage. At greater depth, where the cable is suspended in drilling fluid, seismic interval velocities and attenuation are measured. Vertical P-wave velocities are high ($V_{INT} = 3984 \pm 218$ m s$^{-1}$) and consistent with a strong vertical cluster fabric. Seismic attenuation is high ($Q_{INT} = 75 \pm 12$) and inconsistent with previous observations in ice sheets over this temperature range. The signal level is too low, and the noise level too high, to undertake analysis of englacial fabric variability. However, modelling of P- and S-wave traveltimes and amplitudes with a range of fabric geometries, combined with these measurements, demonstrates the capacity of the DAS method to discriminate englacial fabric distribution. From this pilot study we make a number of recommendations for future experiments aimed at quantifying englacial fabric to improve our understanding of recent ice sheet history.

## 1 Introduction

Uncertainties in sea level rise projections are dominated by our understanding of how the ice sheets will evolve under a warming climate (IPCC, 2013). Satellite observations over the last 40 years have highlighted the rate of ongoing ice sheet change (Shepherd et al., 2018; Rignot et al., 2019). However, knowledge of past ice sheet change is key to reducing uncertainty in projections of the future behaviour of the ice sheets (Siegert et al., 2019). One key region of West Antarctica where consensus on ice sheet history is yet to be reached is the Weddell Sea sector. Contradictory and incompatible recent histories have been proposed. Hillenbrand et al. (2014) reviewed geological observations as well as marine and terrestrial chronological data and proposed a monotonic ice sheet retreat since the Last Glacial Maximum (LGM) to the current grounding line position. However, studies utilising airborne radar surveys (Siegert et al., 2013), ground-based radar and radiocarbon dating (Kingslake et al., 2018) and ice sheet modelling (Bradley et al., 2015) support ice sheet retreat from the LGM to beyond the current grounding line configuration followed by a more recent re-advance. Due to a paucity of absolute dating control, the timing of these changes is also poorly constrained.

Significantly, these contradictory histories imply different stability regimes of the ice sheet in its current configuration, with important implications for a much discussed process, the marine ice sheet instability mechanism, or MISI (Mercer, 1978). MISI describes a feedback mechanism whereby ice streams resting on a reverse bed slope (i.e., deepening upstream) accelerate during retreat. The instability results from an increase in flux at the grounding line as the bed deepens and ice thickness at the grounding line increases as the ice stream retreats. In the Weddell Sea Sector, both the Institute and Möller Ice Streams, for

example, rest on a bed which deepens upstream (Ross et al., 2012). An ice sheet history with monotonic retreat indicates that the current ice sheet configuration may be unstable, with MISI potentially underway. On the contrary, an ice sheet history that includes retreat beyond the current grounding line position with subsequent re-advance to the present-day position demonstrates that the MISI process may be stabilised. For example, in the Amundsen Sea Sector, geodetic observations and modelling results demonstrate that the solid-Earth response to rapid ice loss may result in the stabilisation of retreat (Larour et

al., 2019; Barletta et al., 2018).

Ice flow history is recorded within the structure of the ice column (Conway et al., 1999). Where ice flow is slow, such as at ice rises and divides, the ice structure may be analysed to understand recent ice flow. Ice rises form independent ice flow centres, with characteristic gravity-driven flow originating at the apex (Matsuoka et al., 2015). The stratigraphy of the ice column at the ice divide records this ice flow and may be emulated using numerical ice flow models (e.g., Martin et al., 2009;

Raymond, 1983; Conway et al., 1999). Deviation from the expected structure may be used to infer changes in ice flow (Wearing and Kingslake, 2019; Kingslake et al., 2016; Nereson et al., 1998; Martin et al., 2006).

Ice crystals are strongly anisotropic; The shear strength parallel to the c axis of hexagonal ice crystals is several orders of magnitude greater than perpendicular to it, resulting in c-axis rotation during the deformation of polycrystalline aggregates (Alley, 1988; Duval et al., 1983). Under differential stress, the c-axes rotate towards the compressional and away from tensional

axes, resulting in crystal preferred orientation (CPO), or fabric, that is characteristic of the stress regime (Azuma and Higashi, 1985). At a perfect ice dome where ice deforms by vertical uniaxial compression alone, a cluster fabric with c-axes orientated towards the vertical results (e.g., Azuma et al., 1999; Durand et al., 2007). At ice divides, where a lateral component of tension is present, a vertical girdle fabric also develops, with the c-axes rotated to form a diffuse girdle orthogonal to the direction of extension (e.g., Lipenkov et al., 1989; Bargmann et al., 2012). The nature of fabric at ice domes and ice divides has been

confirmed using laboratory measurements of ice core (Weikusat et al., 2017; Montagnat et al., 2012; Thorsteinsson et al., 1997; Wang et al., 2003). Numerical modelling results of Martin et al. (2009) demonstrate that ice-divide fabrics require at least 4 times the characteristic time to develop, where the characteristic time is the ratio of ice thickness to accumulation rate. The time taken to overprint a pre-existing fabric is poorly constrained and dependent on temperature and strain rate. However, at low strain rates and in the absence of recrystallization it will take significant time to eradicate existing fabric. Therefore, a

complex strain history leads to a complex fabric (Alley, 1988), and observations of fabric as well as structure at ice divides may be used to interpret ice flow history (Brisbourne et al., 2019).

The most widely used method to determine fabric at ice divides is the analysis of ice core. However, recovery of ice cores is logistically complex and expensive. In addition, measurements of fabric in ice cores generally do not include the in situ orientation, as this is lost during retrieval of the core. It is therefore desirable to measure fabric in situ. By extension, it may

also be desirable to measure fabric when an ice core has not been recovered during a drilling campaign, such as following the use of a hot water drill, or a rapid access drill which recovers ice chippings rather than ice core (Rix et al., 2019). Surface geophysics methods and downhole measurements are therefore required. The depth distribution of fabric has previously been investigated using surface seismic reflection methods (Horgan et al., 2011; Horgan et al., 2008; Blankenship and Bentley, 1987; Bentley, 1972) and radar methods (Hargreaves, 1977; Fujita et al., 2006; Doake et al., 2002). With passive seismic

methods, deriving the depth distribution of fabric is not always possible (Smith et al., 2017). Additional constraints on fabric type and distribution within the ice column may be achieved using measurements at depth within the ice column. Direct access to the deep ice column is logistically difficult, expensive and risky. However, large-scale ice core or hot water drilling experiments potentially provide access to the subsurface as a by-product of the primary objectives of the experiment. Although it is feasible to deploy 3-component borehole seismometers if holes are sufficiently wide, these systems are expensive and,

dependent on the number of instruments deployed, provide measurements at a limited number of depths within the ice column

(Lutz et al., 2020). To compensate for this limitation, Diez et al. (2015) moved a single downhole geophone incrementally up through an ice core borehole with repeat surface shooting to produce a vertical seismic profile. Comparing these results with radar and ice core analysis results, they identified both strong cluster and girdle fabric at an ice divide site and demonstrated consistency between the methods. Downhole geophysical measurements remain sparse in glacial settings (e.g., Obbard et al., 2011; Hubbard et al., 2020; Raymond, 1971; Hubbard et al., 1998; Roberson and Hubbard, 2010), reflecting the logistical challenges of such methods. Again, such systems require access to a sufficiently wide and uniform borehole without the risk of hole closure, which is often not the case with hot water drilling in dynamic ice flow regimes for example.

However, following ice core recovery or hot water drill access it is now common practice to deploy fibre optic cables downhole to measure temperature profiles within the ice column using the Distributed Temperature Sensing (DTS) method (Ukil et al., 2012). Distributed acoustic sensing (DAS) provides an opportunistic method to exploit existing infrastructure, enabling downhole measurements at little extra cost and minimal risk. Furthermore, measurements using DAS may be acquired throughout the ice column, are capable of sub-metre sample intervals, and if the fibre optic cable remains in situ, are readily repeatable to allow analysis of temporal variation. DAS involves measuring the Rayleigh backscattered returns along optical fibre cables. Time-varying strain of the fibre, such as that from passing seismic waves, is derived from changes in the phase difference in backscattered light from closely spaced points along the fibre. The phase-lags measured by an interrogator are used to reconstruct the seismic record (Hartog, 2017). The distributed nature of the Rayleigh scatterers within the fibre emulates a continuous geophone string. Such records are analogous to standard seismic methods, but unlike with standard geophone sensors, measuring strain rate rather than velocity. Where a fibre optic cable is deployed down a borehole, the DAS method can be used to derive a vertical seismic profile (VSP). Standard seismic processing methods are then applied to calculate, for example, seismic interval velocities from the traveltime through layers of known thickness. Fibre optic cables are termed single-mode (with a light carrying core of 8-10 μm) or multi-mode (with a light carrying core of > 50 μm). Due to multiple paths within the fibre, multi-mode fibre results in signal dispersion, reduced range and lower bandwidth. However, multi-mode fibre produces a higher threshold for nonlinearities and allows more power to be applied to the fibre, resulting in stronger backscatter signals, and is therefore preferred by some manufacturers for certain applications, such as distributed temperature sensing (DTS). The backscattered signal used in DAS is stronger than DTS and therefore single-mode systems are commonly used for DAS measurements. A fibre optic cable may also utilise a "bend" whereby two lengths of fibre may be spliced together to form an outward and return fibre within the same cable, thereby allowing two measurements to be made at each point along a cable with a single interrogator unit. Although measurements may be reported with a channel spacing of 0.25 or 1 m for example, strain is measured over a finite length of fibre, or spatial distance, referred to as the gauge length, and is typically 3 or 10 m. The gauge length is therefore effectively a moving spatial average filter applied to measurements at the channel spacing.

To our knowledge, DAS technology has been deployed only twice previously in glaciological studies, and never in Antarctica. Booth et al. (2020) presented results from a VSP with borehole deployment of DAS on Store Glacier, West Greenland. Measurements of interval seismic velocity and attenuation, as well as sub-glacial reflections demonstrated the potential of the DAS method in glaciological settings. Using a deployment of fibre optic cable on an alpine glacier surface, Walter et al. (2020) demonstrated the capacity of the method to record naturally occurring glacial seismicity, and demonstrated the capacity for these observations to locate icequakes and delineate physical properties of the glacier and its bed. We present results from the first field deployment of DAS in an Antarctic borehole. We occupied a recent ice core site to evaluate the potential of DAS technology to constrain the seismic structure of the ice column at an Antarctic ice rise with the aim of improving understanding of ice sheet history. We demonstrate the capability of the method in such scenarios with field observations, highlight the potential for the method to discriminate englacial structure through modelling, and make recommendations for future deployments in similar situations.

## 2 Field setting and data acquisition

Skytrain Ice Rise (SIR) in the Weddell Sea region of West Antarctica is an independent ice flow centre adjacent to the Filchner-Ronne Ice Shelf, bounded by the fast-flowing Rutford and Institute Ice Streams, Constellation and Hercules Inlets, and the southern end of the Ellsworth-Whitmore Mountains (Fig. 1a,b). In 2018/19, as part of the WACSWAIN Project (Mulvaney et al., 2021), an ice core was recovered to bedrock at 651 m depth close to the highest point of the ice rise (Fig. 1b). The experiment site is directly above a Raymond Bump visible in radar profile data (Mulvaney et al., 2021), a feature observed within the ice column that is indicative of long-term stable ice flow. Surface ice flow speed is < 10 m a⁻¹ (Rignot et al., 2011). Following extraction of the ice core, three cables were deployed in the borehole: (1) a thermistor string to a depth of 635 m with instruments at known depths (635, 585, 535, 385 and 235 m); (2) a PT-100 platinum resistance thermometer to 95 m depth, and (3) a multi-mode fibre optic cable to 595 m depth, principally for DTS. The three cables were joined at 5 m intervals. The borehole remained open due to a combination of slow ice flow at the ice divide and the presence of drilling fluid in approximately the lower half of the borehole (exact depth unknown).

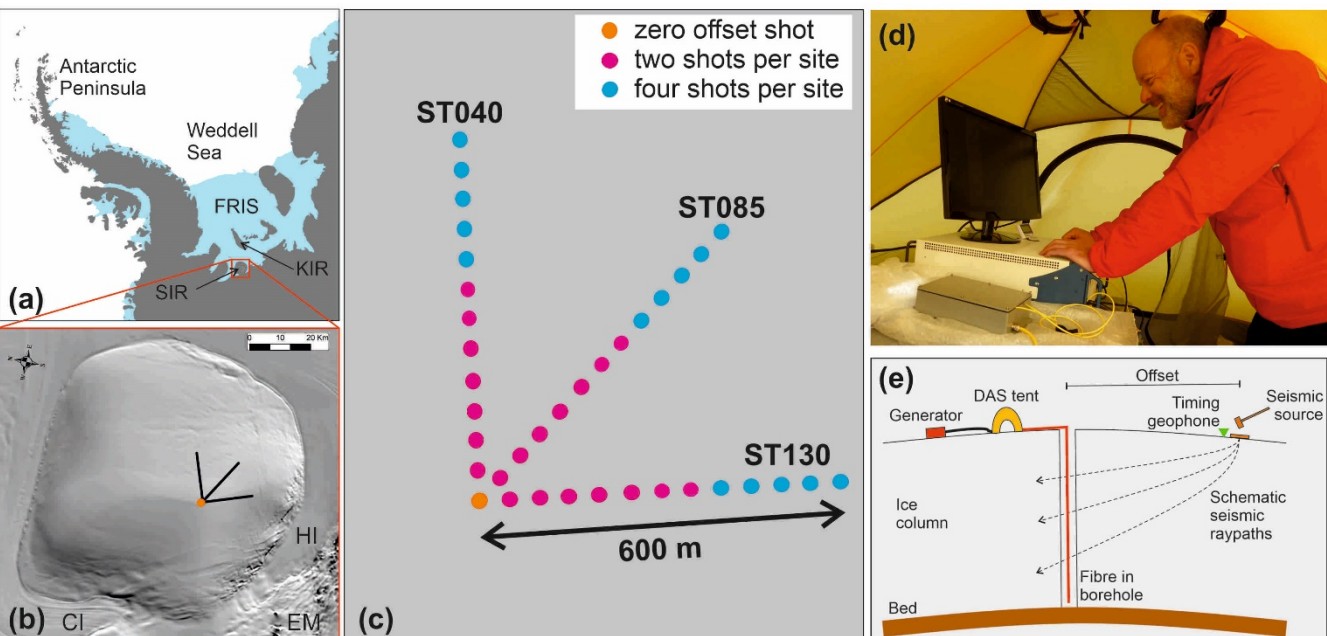

**Figure 1: (a) Location of the experiment (red dot) on SIR in West Antarctica. FRIS – Filchner-Ronne Ice Shelf; KIR – Korff Ice Rise. (b) Orientation of walkaway hammer and plate seismic lines on SIR (not to scale). The background is MODIS imagery (Scambos et al., 2007). HI – Hercules Inlet; CI – Constellation inlet; EM – Ellsworth-Whitmore Mountains. (c) Scale map of shot locations with respect to the borehole at 79° 44.5'S, 078° 32.7'W (orange dot). Line names reference the orientation with respect to magnetic north. (d) DAS interrogator in a mountain tent. (e) Schematic of acquisition illustrating the key components of the field setup.**

In January 2020 the site was revisited to conduct a VSP experiment centred on the borehole using a Silixa iDAS™ interrogator unit with GPS timing. A 1.0 kVA petrol generator was used to power the interrogator unit. The interrogator was housed in a tent to protect it from the wind and maintain a working temperature. Data were recorded at 8000 Hz sampling frequency. Strain rate measurements were made at 1 m intervals along the fibre but with an effective resolution of 10 m due to the "gauge length" or strain rate averaging process (Dean et al., 2017). The fibre optic cable includes a bend at 595 m, where an identical length of fibre within the cable is spliced and returned to the surface, enabling two measurements to be made at each depth. The DAS method measures longitudinal strain. Therefore, with the fibre optic cable vertical, only the vertical strain component of seismic

150 waves propagating past the fibre is measured. Consequently, the cable is sensitive to vertically propagating P-waves but not vertically propagating S-waves.

The acquisition geometry is presented in Figure 1c. With the aim of discriminating anisotropic structure within the ice column, three walkaway profiles were acquired at 45° to one another. Line names reference the orientation with respect to magnetic north. A 4.5 kg sledgehammer with a rigid polyethylene impact plate stamped into the snow surface acted as the energy source.

155 Hammer impact times were derived from a continuous 1000 Hz record of a GPS-synchronized 3-component surface geophone deployed adjacent to the hammer plate (Fig. 1e). Profiles were acquired out to 600 m offset from the borehole with hammer blows at 50 m intervals. Two hammer blows were made at each shot point out to 350 m and four hammer blows from 400 m and beyond. The interrogator unit recorded in continuous mode with shot gathers subsequently extracted using times derived from impulsive arrivals on the surface geophone recording made adjacent to the hammer plate.

160 **3 Observations and data processing**

Figure 2 presents example VSP records for individual hammer blows at a range of offsets along line ST130. Data are "folded" about the bend in the cable, resulting in a stack of two traces for each hammer blow. Figure 2a presents the zero offset or "checkshot" (i.e., source at the top of the borehole) bandpass filtered at 2-140 Hz. The downwards propagating P-wave, the primary ice base reflection and the surface multiple are all visible. The primary downwards-propagating P-wave arrival is

165 disrupted at around 350 m depth. A mixture of upgoing and downgoing energy is observed at 100-300 m depth following the primary P-wave arrival, producing a diamond-shaped signal pattern on a depth-time plot. From zero to 150 ms, prior to the primary downgoing P-wave arrival, a horizontal signal is observed in Figure 2a. A signal that plots horizontally in a VSP record indicates an arrival coincident in time along the length of the fibre optic cable. This is however unlikely in this situation and is most likely due to vibration of the DAS interrogator unit, a result of wind and generator noise at the surface.

170

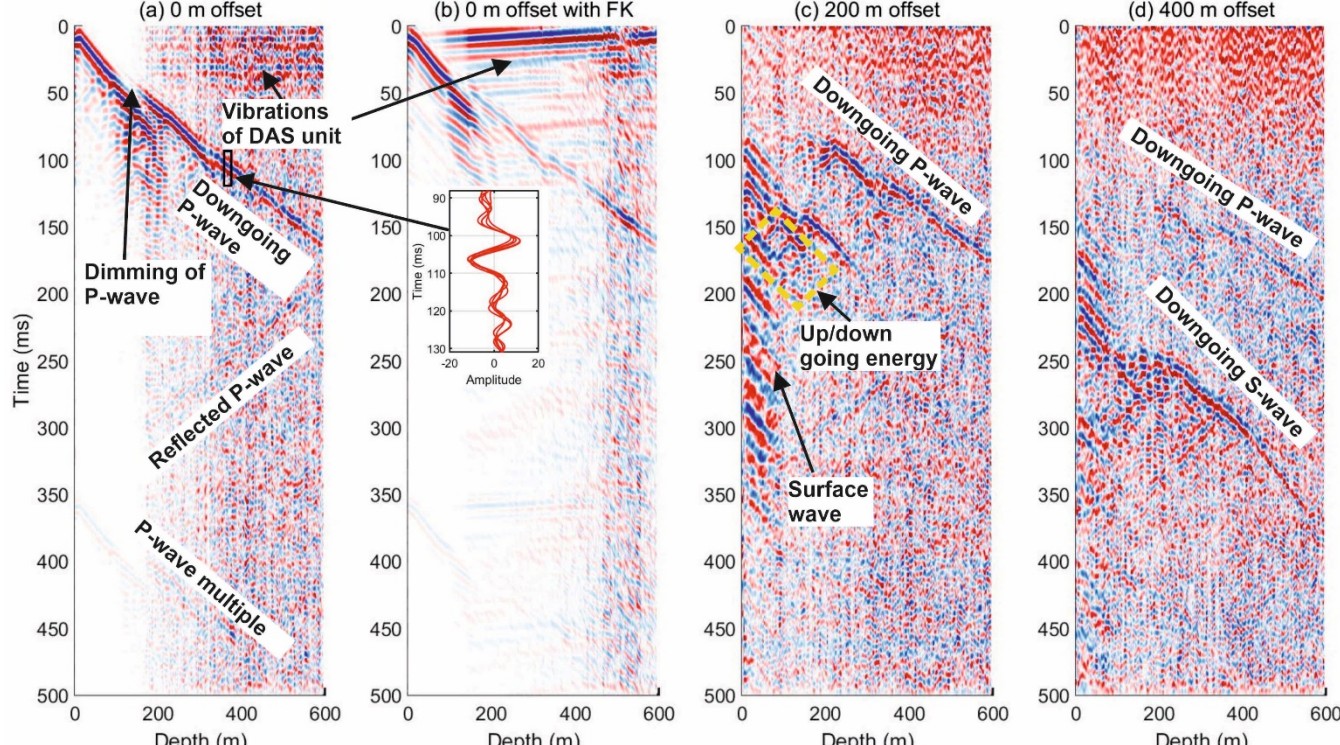

**Figure 2: Example DAS VSPs along line ST130 with signal and noise labelled. Traces are normalised at each depth to highlight coherent arrivals. This also results in the dimming of signals over depths with high amplitude arrivals dominate. (a) Primary, reflected and surface-multiple P-waves at zero offset with a bandpass filter of 2-140 Hz. Vibrations from the generator and wind noise of the DAS interrogator appear as horizontal lines in these sections. The apparent dimming of the P-wave at 50 ms is due to the high amplitudes of the following signals; (b) Downgoing P-wave energy at zero offset isolated using an F-K filter and adaptive deconvolution to remove upgoing energy and reduce coherent noise. Some upgoing energy is preserved due to the taper used on the FK filter. Inset: Example waveforms of five downgoing P-waves at 100 ms and 350 m depth, indicated by the black box in Figure 2a (no FK applied). (c) Source at 200 m offset with a bandpass filter of 2-140 Hz. The characteristic diamond-pattern is highlighted by yellow-dash boxes; (d) Both P-wave and S-wave arrivals are visible at 400 m source offset with bandpass filter of 2-140 Hz applied.**

Figure 2b presents the checkshot record as presented in Figure 2a, but with a frequency-wavenumber (FK) filter applied to suppress upward propagating seismic energy (negative wavenumber energy removed, 2-140 Hz bandpass) and with an adaptive deconvolution filter (Griffiths et al., 1977) applied to remove coherent noise. The upgoing energy is removed, but the primary P-wave energy remains discontinuous with the wavetrain disrupted. The combined up-downgoing signal at approximately 100-300 m depth is suppressed by the combination of the FK and adaptive deconvolution filters. However, some upgoing energy persists at depths of less than 300 m. Over a shorter time interval and shallower depth range, Figure 2c presents details of the VSP record acquired with a shot offset of 200 m on line ST130. A 2-140 Hz bandpass filter is applied. Again, the downgoing P-wave energy is clear, with a delay due to the source offset from the fibre. Relatively high-amplitude but low-frequency energy is present to depths of 100 m. The characteristic diamond pattern is again observed, to a depth of up to 250 m, indicating both up and down-going energy. Figure 2d presents the VSP with a shot offset of 400 m and a 2-140 Hz bandpass filter applied. At this offset, seismic waves propagate more horizontally. Therefore, downgoing S-wave energy dominates the record with downgoing P-wave energy only visible beyond approximately 300 m depth.

### 3.1 Noise sources

Figure 3a presents a single VSP from a 200 m offset shot along line ST130. The downgoing P-wave is visible at depths below 200 m and the downgoing S-wave at 150 to 230 m. A low frequency signal is also visible at times greater than 150 ms and from the surface to a depth of 100 m. As described above, significant noise is present on the records at depths of less than 300 m. This takes the form of "diamond shapes" indicating a combination of upgoing and downgoing energy. The most obvious source of signals of this type is englacial reflections of downgoing energy. However, a number of factors indicate a different noise source of such signals. The firn column consists of snow compacting and metamorphosing and densifying under its own weight to form solid ice, with a pore close off depth at SIR of 56 m (Mulvaney et al., 2021). Below this depth the ice column likely varies very little at scales sensitive to the seismic wavelengths analysed here ($\lambda \sim 36$ m) and the seismic velocity is predominantly controlled by temperature, which varies slowly. The low temperatures at SIR (-26.0° C at 10 m depth (Mulvaney et al., 2021)) preclude ice layers in the firn. Therefore, there is no likely source of layering which could result in seismic reflections of any significance from within the ice column. Similarly, the stress regime precludes englacial crevassing. No structures are observed in the radar profile collected across SIR (Mulvaney et al., 2021, Fig. 3).

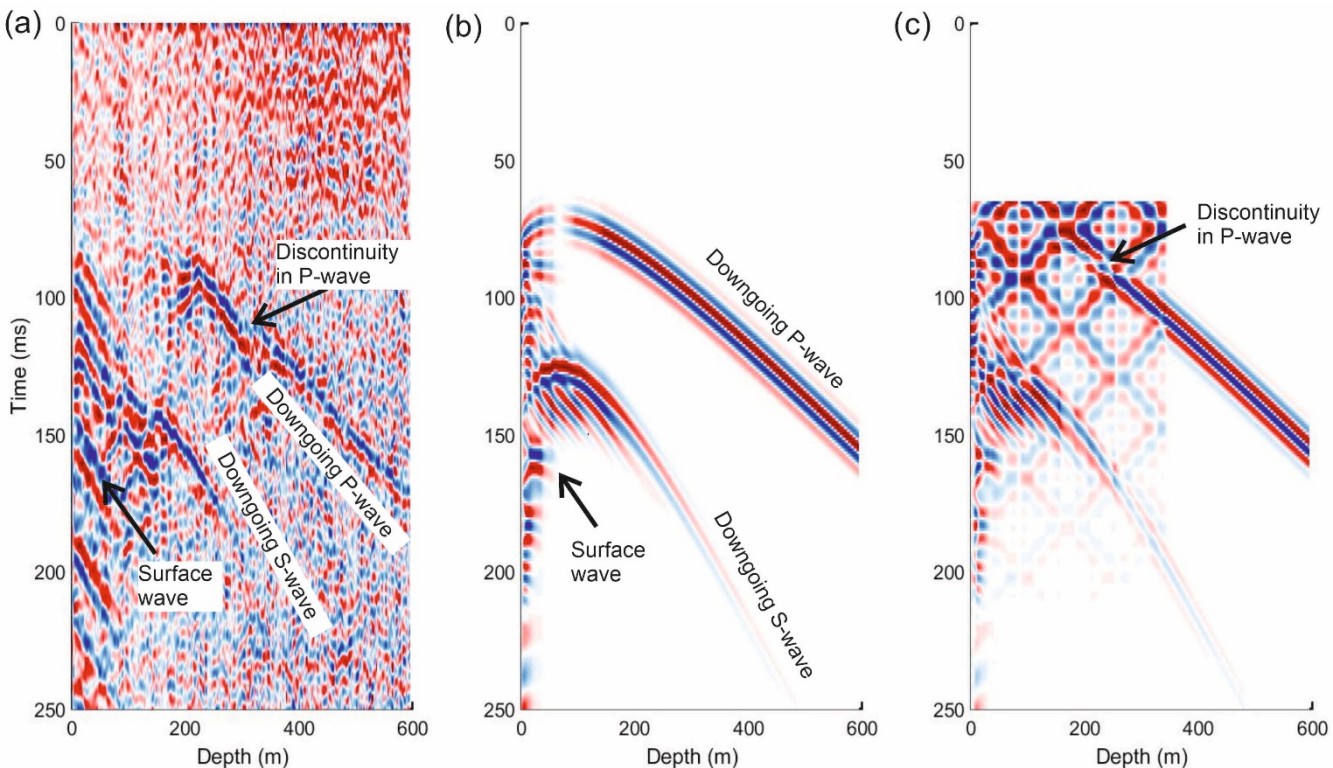

**Figure 3: Observed and synthetic VSPs with a shot offset of 200 m from the top of the borehole along line ST130. Traces are normalised at each depth. (a) Observed VSP with a 2-140 Hz bandpass filter applied; (b) Synthetic VSP with the principal arrivals labelled; (c) Synthetic VSP summed with simple harmonic motion of a vibrating string fixed at both ends (0 m and 340 m).**

We therefore propose a different source for the diamond-shaped signal. As stated above, the hole remains open. The fibre optic cable is unlikely to be frozen to the borehole (R. Mulvaney, Pers. Comm.). This is in contrast to the experiment of Booth et al. (2020) where the borehole closed-up within a few days. Prior to close-up of the borehole, Booth et al. (2020) observed a similar diamond-shaped signal pattern in the VSP data. We therefore attribute this noise to the fibre being unclamped in the hole and hanging in air. At depths where the drilling fluid is present (below around 300 m) the fibre optic cable sits in the fluid and couples to the walls of the hole via the fluid. At these depths we can expect lower noise levels due to damping of vibrations of

the cable and improved signal coupling. This is indeed the case in our data where we see a transition at around 300 m depth. This depth also coincides with a discontinuity in the primary P-wave arrival, consistently observed over a range of shot offsets and azimuths. The onset of diamond-shaped noise is coincident with the first P-wave arrival, in this case at around 50 ms, and we therefore attribute noise of this form to the excitation of vibrations on the cables by seismic energy passing the borehole.

To test this hypothesis we construct full-waveform synthetics to simulate VSP records with added coherent noise due to cable resonance. Figure 3b shows a synthetic VSP simulating a 200 m offset shot into a 600 m downhole fibre optic cable. We use SPECFEM2D current source (Tromp et al., 2008) published under the CeCILL V2 license. We convert the synthetic seismogram from velocity to strain rate to emulate DAS measurements. We assume the seismic velocity profile as measured at the adjacent Korff Ice Rise (Brisbourne et al., 2019). In this synthetic VSP (Fig. 3b), clear P- and S-wave energy is observed, consistent with the observations. The low frequency surface waves in the synthetic seismogram are coincident with a low frequency arrival in the observations (Fig. 3a). We hypothesise that as the surface wave traverses the borehole it excites a mode of vibration on the fibre that propagates its length, resulting in a consistent noise signature across the range of shots. We argue that this signal is not a depth-observation of the surface wave due to slope of the arrival in Figure 2c. A surface wave traversing the fibre would appear coincident along the fibre, i.e. with a horizontal nature in the VSP.

In Figure 3c we present the sum of the synthetic seismogram and a synthetic noise signal generated assuming simple harmonic motion (SHM) on a string of length 340 m fixed at both ends and with nodes at 85 m intervals. These parameters are set at values that produce a reasonable simulation of the observations and are not intended to match the conditions down the borehole, which are poorly constrained. We scale the amplitude of the SHM to produce a representative match to the data and apply exponential decay with time to this signal. We initiate the synthetic SHM at the arrival time of the P-wave at the borehole. The SHM produces the characteristic diamond-shaped signal as observed in the VSPs. The summation of the synthetic seismogram and noise also produces nulls in the resultant direct wavetrain, similar to those observed in the data. The synthetic profile is not an exact representation of the observations. However, similarities with the observations indicate that the most likely source of this diamond-shaped noise is vibrations on the cables suspended in air, initiated by the propagation of the seismic signal past the borehole.

Unfortunately, the effect of cable resonance in the open parts of the borehole (surface to ~350 m depth) render it difficult to use data from this section of the cable in an analysis of seismic properties of the ice column. Hence, we only expect reliable results from the cable in the deeper parts of the borehole (350 to 550 m depth). To improve signal to noise levels for further data analysis the "folded" shot records are stacked to combine all hammer blows at each shot point.

## 4 Data analysis and results

### 4.1 Seismic velocity

Prior to velocity analysis we apply an FK filter, passing 2-140 Hz and removing negative wavenumbers to reduce upgoing energy. We also apply an adaptive deconvolution filter to remove coherent noise. Traveltime picks are made at the peak of the first arrival rather than the first break. This method is valid where only relative traveltimes are required, for example to calculate interval velocities. We use a sliding window of 50 m length to determine interval velocities along the three azimuths. When calculating seismic interval velocities, the thickness of the interval selected depends on the trade-off between resolution and noise levels. Within each 50 m window we determine the best linear fit to the traveltime picks and calculate the interval velocity from the gradient. Velocity uncertainties are calculated using the one standard deviation uncertainty of the linear traveltime fit over each 50 m sliding window. The range of velocities over each window is calculated by perturbing the interval traveltime by plus and minus two standard deviations to emulate the maximum possible range of traveltime intervals. As depths

are relative, the error in interval length is negligible. Figure 4 presents interval velocities from the 0, 50 and 100 m offset shots.
Table 4 presents mean interval velocities between 380 and 540 m shots over the three azimuths and offsets out to 100 m. We
select the upper limit of this interval to exclude the harmonic noise in the upper half of the ice column and the discontinuity in
the P-wave arrival at approximately 300 m depth. Below 540 m depth, the sliding window length precludes further
measurements. Although the interval velocities between 200 and 300 m are consistent, the interaction with the harmonic noise
leads us to question the reliability and we do not evaluate this interval any further. To convert from apparent to true velocity
we calculate the incidence angle of the wavefront at the vertical fibre optic cable by ray tracing to the centre of the receiver
interval with an isotropic velocity model based on measurements at the nearby Korff Ice Rise (Brisbourne et al., 2019; Guest
and Kendall, 1993).

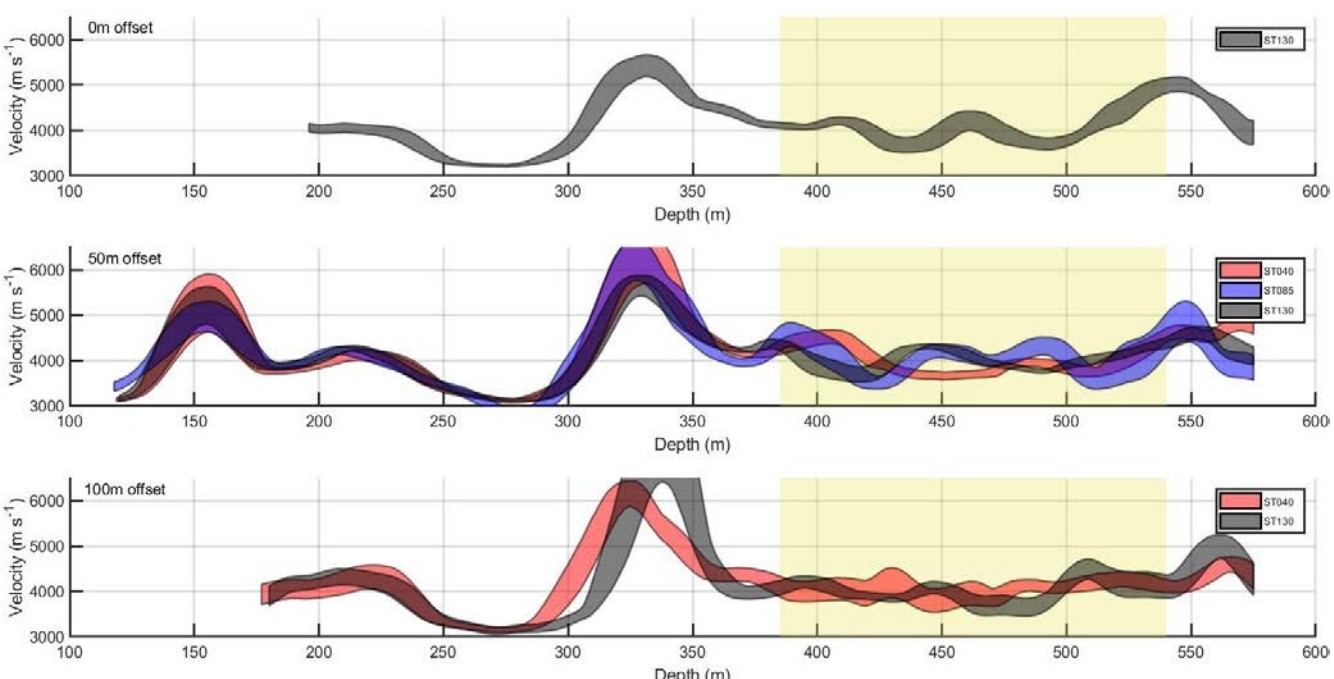

**Figure 4: Vertical P-wave velocity from 0 m (upper plot), 50 m (middle plot) and 100 m offset (lower plot) shots calculated using a
linear fit to 50 m sliding windows. Velocities are converted from the apparent velocity assuming incidence angles at the midpoint of
the respective sliding window following ray tracing through an isotropic model. The width of the fitted curves represents the velocity
uncertainty calculated from the traveltime uncertainty over each sliding window. The yellow box represents the depth range over
which mean interval velocities are presented in Table 1 (380 to 540 m depth).**

The mean velocity of 3984 ± 218 m s$^{-1}$ is high but comparable to measurements of Booth et al. (2020) in the deepest part of
the ice column of Store Glacier, Greenland, and Diez et al. (2015) at the EDML ice core site in Dronning Maud Land,
Antarctica. Although uncertainties are large (Table 1) they are comparable to uncertainties in VSP measurements at depth in
ice of Diez et al. (2015) and Booth et al. (2020). Both of these previous studies attribute the high velocities to a fabric with
aligned vertical c-axes. A vertical cluster is consistent with the fabric development expected at ice divides using a numerical
model (Martin et al., 2009), and the measurements of Diez et al. (2015) using both seismic VSP data and COF measured in ice
core. Uncertainties in the velocity measurement preclude interpretation of fabric from depth or azimuthal variation in P-wave
velocity. The signal to noise level is too low to derive reliable S-wave velocities.

**Table 1** Vertical interval velocities measured between 380 and 540 m depth. Uncertainties are one standard deviation.

| Line | Shot offset (m) | $V_{INT}$ (m s$^{-1}$) |
|---|---|---|
| ST040 | 50 | 3996 ± 298 |
| ST040 | 100 | 3983 ± 107 |
| ST085 | 50 | 4079 ± 271 |
| ST085 | 100 | - |
| ST130 | 0 | 3952 ± 188 |
| ST130 | 50 | 3989 ± 215 |
| ST130 | 100 | 3902 ± 171 |
| **Mean** | | **3984 ± 218** |

## 4.2 Seismic attenuation

The spectral ratio method (Bath, 1974) is applied to the direct P-wave arrivals to determine seismic attenuation along vertical propagation paths within the ice column (Figure 5). The natural logarithm of the spectral ratio is derived;

$$ln\left|\frac{A_1(f)}{A_2(f)}\right| = const. + \pi\frac{\delta t}{Q}f \tag{1}$$

Where $f$ is the frequency of the seismic wavelet and $\delta t$ is the measurement traveltime interval. The slope of the spectral ratio $m = \pi\frac{\delta t}{Q}$ yields the seismic quality factor, $Q$.

We apply the spectral ratio method across a single depth interval from 380 to 540 m, consistent with the interval velocity measurements. We use the zero-offset shots to analyse vertical raypaths as a strong dependence of attenuation on crystal orientation has previously been demonstrated (Oguro et al., 1982). To aid phase picking, traces are bandpass filtered at 2-140 Hz. Traces are then windowed around the primary P-wave arrivals. DAS measurements of strain rate implicitly average over a gauge length of 10 m (defined by the DAS hardware). In addition, at SIR, little change is expected in the ice column over a 10 m depth range. Therefore, to improve the signal to noise ratio, traces are stacked over a 10 m window at the respective depths. To remove noisy traces and compensate for the discontinuous nature of the wavetrain, prior to stacking we perform a cross correlation to evaluate each trace against the central trace of the respective interval. Traces with a correlation coefficient < 0.95 are discarded, removing 45% of traces (Fig. 5a-c). To calculate the uncertainty in the gradient measurements the 95% confidence limits of the stacked traces are used, analysing waveforms at the extremes of this envelope (Fig. 5a,b) to calculate a range of spectral ratios and therefore gradients. In addition, the uncertainty in the gradient fit and traveltimes (±1 ms) are used to determine the uncertainty in Q. A mean interval Q of 75 ± 12 for this lower section of the ice column is measured. The uncertainties are consistent with previous studies that use the spectral ratio method (Dasgupta and Clark, 1998; Peters et al., 2012). The temperature measured over this depth range in the ice column is -20° to -17° C (Mulvaney et al., 2021).

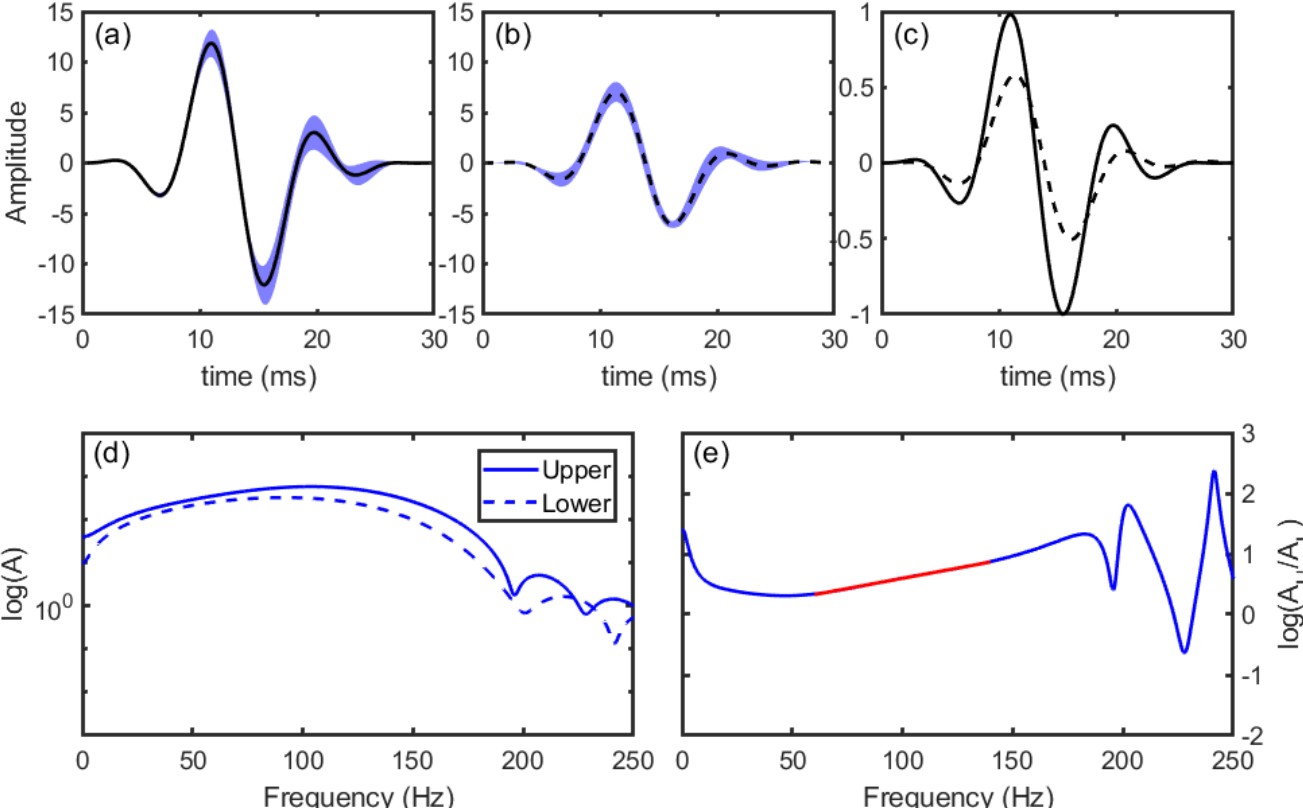

**Figure 5: Use of the spectral ratio method to determine seismic attenuation. Traces are stacked around the direct P-wave arrivals over a 10 m interval. The cross-correlation with the central trace in the respective depth window is calculated and only traces with a coefficient > 0.95 are used. (a) Upper stacked trace (380 to 390 m; solid line) and (b) Lower stacked trace (530 to 540 m; dashed line), both with 95% confidence bounds calculated from the summed traces in blue; (c) Upper (solid line) and lower (dashed line) stacked traces with normalized amplitude; (d) Amplitude spectra of the upper and lower stacked waveforms. (e) Log spectral ratio with the frequency range over which the gradient is measured in red (60-140 Hz).**

## 5 Evaluating the potential of DAS VSP for ice column fabric discrimination using synthetic traveltimes and amplitudes

Our observations of P- and S-waves, both direct and reflected, and at a range of offsets and azimuths, demonstrate the potential for DAS type measurements at Antarctic ice rises. We therefore investigate the capacity of DAS measurements to discriminate ice column fabric at ice rises to guide future experiments investigating ice sheet history through fabric evolution (e.g., Brisbourne et al., 2019). We use the software package ATRAK (Guest and Kendall, 1993), based on asymptotic ray theory (Kendall and Thomson, 1989), to calculate traveltimes and amplitudes of direct P- and S-wave arrivals from a range of offsets and azimuths. We used an isotropic source with identical P, SV and SH amplitudes. An isotropic seismic velocity profile is used in the firn, derived from a previous seismic refraction experiment at Korff Ice Rise (Brisbourne et al., 2019). We test three models of fabric at ice rises: (1) Isotropic; (2) A model with an isotropic layer over a layer with a strong cluster fabric at depth. Critically, radar methods with collocated source and receiver are insensitive to vertical cluster (or azimuthally invariant) fabrics, and seismic methods provide an approach to quantify any azimuthally invariant fabric (Brisbourne et al., 2019); (3) A model based on the ice rise fabric predicted by Martin et al. (2009) with a girdle fabric in the upper ice column (80 to 230 m depth) and a cluster fabric at depth (230 m to the ice base). The strength of the girdle fabric and fabric depth transition is based on that reported from Korff Ice Rise by Brisbourne et al. (2019). With a vertical cable, DAS is only sensitive to the vertical component of strain. Therefore, we derive amplitudes from the vertical component only. Displacement amplitudes computed with ATRAK are relative only and assume an isotropic source function with perfect receiver coupling.

Figure 6 presents modelled P- and SV-wave traveltimes for a shot offset of 400 m and the vertical component amplitudes for shot offsets of 200 and 400 m for the three models of fabric. In the isotropic case (Fig. 6a), as expected there is no azimuthal variation in traveltime or amplitude. Amplitudes vary with depth, with the relationship dependent on shot offset. This is a result of both raypath geometry due to the strong velocity gradient in the firn and the limitation of the downhole fibre DAS method, which measures only vertical strain rate.

The cluster-only model (Fig. 6b) exhibits no azimuthal variation in traveltime or amplitude. However, in comparison with the isotropic case a very strong depth dependence of SV-wave amplitude is observed. The depth of the peak in SV-wave amplitude is dependent on the offset of the shot: shots at progressively greater offset produce peaks in SV-wave amplitude at progressively greater depth. This is a result of the focussing of S-wave energy by the strong cluster fabric in what is a homogeneous but anisotropic layer. The depth of the anisotropic layer will also control the depth distribution of amplitudes. This amplitude

variation is a diagnostic feature that could be quantified in data acquired with a seismic source designed specifically to produce S-waves, such as that of Lutz et al. (2020). Relatively strong S-wave arrivals are observed in the far offset data from SIR (Fig. 2d) although no attempt is made to quantify amplitude variation due to the low signal to noise level. For the extreme form of this fabric, a single maximum, this focussing produces a triplication or folding in the wavefront, which greatly increases the amplitude at depth (results not presented here). Direct observation of a triplication would only be feasible with very high

quality measurements and the strong fabrics, and is likely to be beyond the experimental capability (Baird et al., 2017).

The presence of a girdle fabric would in general result in azimuthal variation in traveltimes. However, in our model the effect on P-wave traveltimes is small (Fig. 6c). For example, the azimuthal variation in P-wave traveltimes is less than 4 ms with either a 200 or 400 m source offset. At 200 m offset the azimuthal variation in SV-wave traveltimes is again small, at around 4 ms. However, with a source offset of 400 m, we see an azimuthal variation in SV-wave traveltime of over 10 ms, which is

likely above the measurement uncertainty with high quality data and therefore a diagnostic parameter. In the same model, SV-wave amplitudes vary with azimuth by up to 50% at 450 m depth and are therefore a robust indicator of fabric if reliable amplitude measurements are available. The azimuthal variation in amplitudes displays maxima at 90° intervals, i.e. a four-fold symmetry. There is also a strong depth dependence of the amplitude maxima, which increases with increasing shot offset. Depending on the fabric geometry, variation in traveltimes and amplitudes with depth and azimuth are diagnostic features and

highlight the viability of the DAS method to constrain englacial fabric.

In Figure 7 we present synthetic VSP gathers for the two end-member models presented in Figure 6a and 6c respectively. We use P, SV and SH traveltimes and amplitudes from the anisotropic ray tracing convolved with a 100 Hz centre frequency synthetic wavelet and its Hilbert transform. These gathers display the features observed in Figure 6, such as the higher amplitude S-waves at depth in the isotropic model (Fig. 7a), azimuthal variation in S-wave amplitude (Fig. 7b), and a peak in

S-wave amplitude at 450 m depth with a 400 m offset source (Fig. 7b). Figure 7c presents details of the S-wave arrival at 250 to 350 m depth in the anisotropic model. The synthetic results indicate that, despite the DAS VSP method measuring only the vertical component of the strain rate, along the 45° orientation in the fully anisotropic model we would observe split S-waves beyond 280 m using a 400m offset source. This phenomenon, shear wave splitting, is a key diagnostic method for quantifying anisotropy using the seismic method (Savage, 1999). In this case, the later arriving S-wave originates with a horizontally

polarized (SH) source. The girdle fabric must therefore impart a rotation of the waveform resulting in a vertical component of particle motion at the borehole. We see no reliable indication of shear wave splitting in our data from SIR. This modelling result is reliant on an isotropic source with identical P, SV and SH amplitudes. Whether it is possible to generate a seismic source that satisfies these requirements warrants further investigation. In subsequent studies, if shear wave splitting is apparent in the data, it will be pertinent to extend plots like those in Figure 6 to include the SH amplitudes and traveltimes.


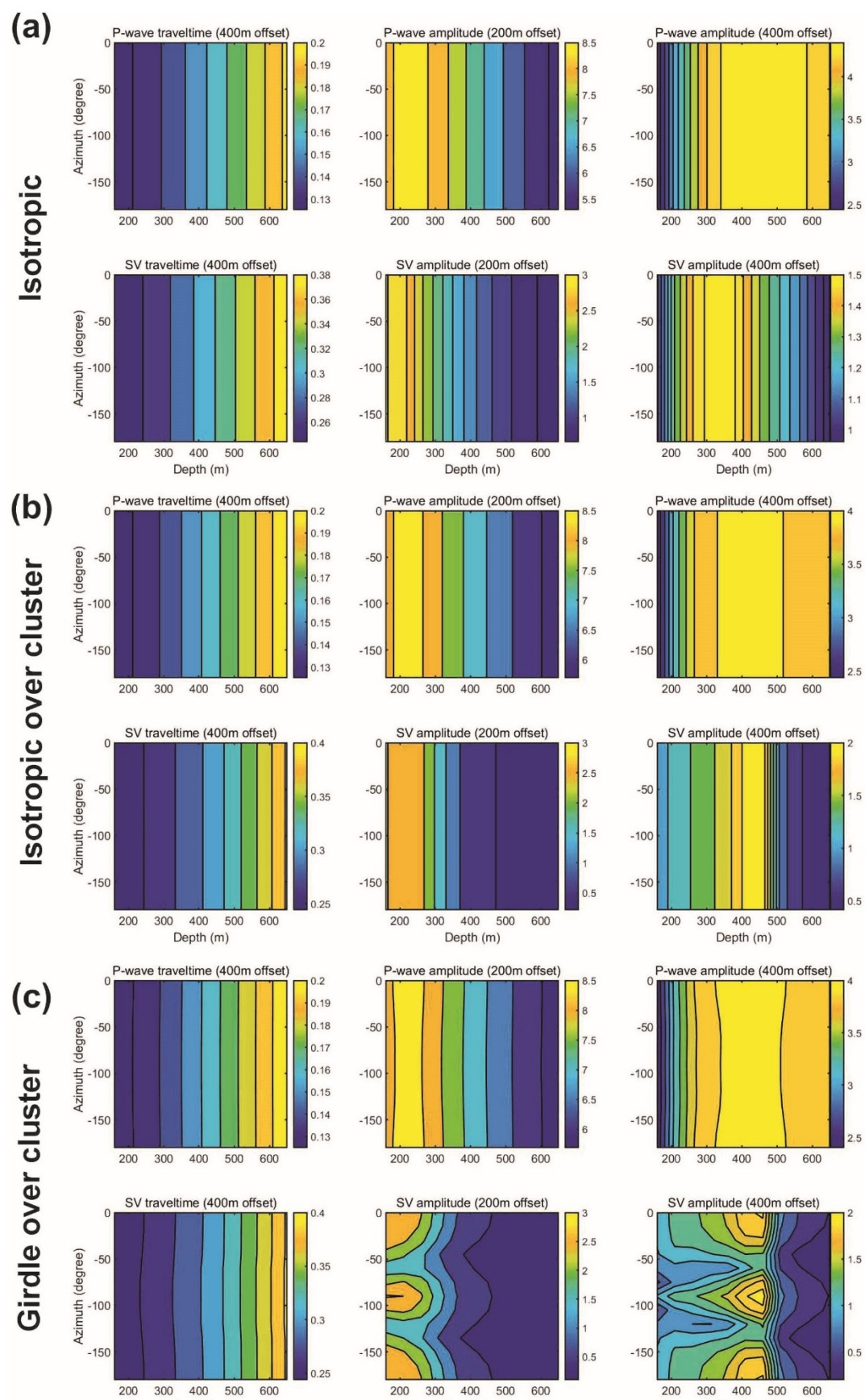

**Figure 6: Modelled P-wave and S-wave traveltime (seconds) and vertical amplitude variation with depth and azimuth (a) isotropic and (b) isotropic with cluster fabric below 230 m, (c) theoretical steady-state ice divide fabric of girdle above strong cluster (Martin et al., 2009). Firn is isotropic in all models and where present the transition to cluster fabric is at 230 m depth. Amplitudes are calculated using the vertical component, consistent with the DAS VSP method.**

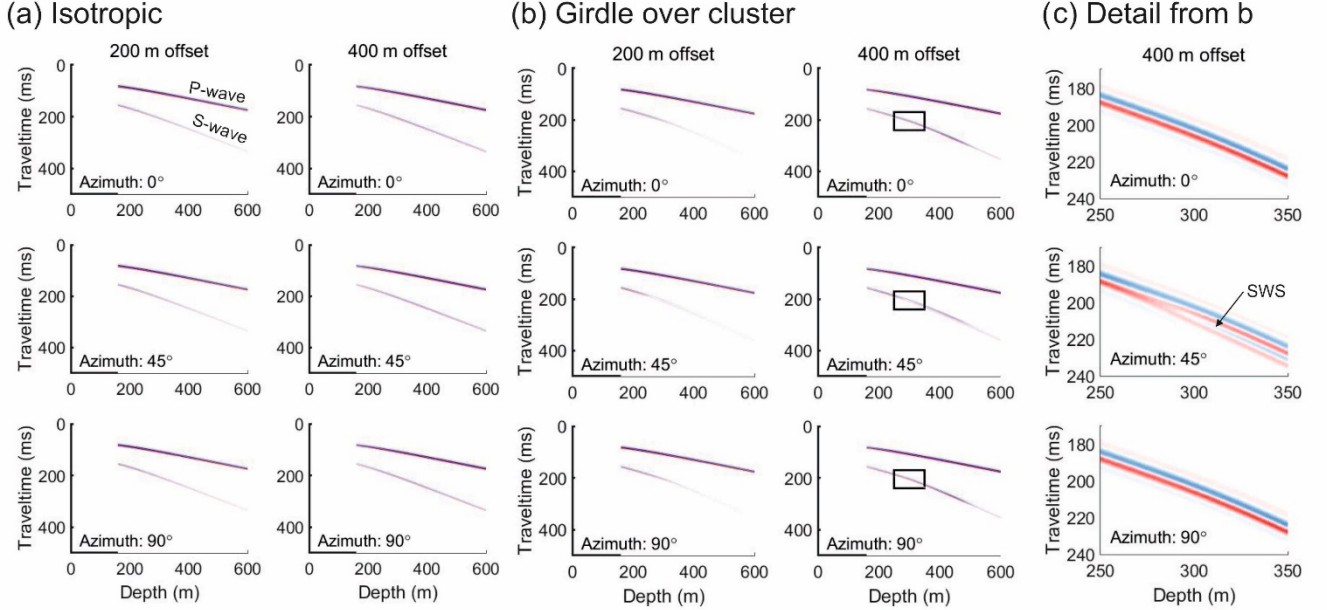

**Figure 7: Synthetic VSP gathers at shot offsets of 200 and 400 m along three azimuths with 45° rotation. (a) isotropic velocity model; (b) theoretical steady-state ice divide fabric of girdle above strong cluster (Martin et al., 2009). Black rectangles highlight the range of arrivals presented in detail in 7c. (c) Detail of S-wave arrivals at 400 m offset in the windows highlighted by the black rectangle in 7b. S-wave arrival showing shear wave splitting (labelled SWS) along the 45° azimuth. Firn is isotropic in all models and where present the transition to cluster fabric is at 230 m depth. Amplitudes are calculated using the vertical component, consistent with the DAS VSP method. All sample values are normalised to the maximum value of both models. The anisotropic case highlights peaks in the S-wave amplitude at around 450m with a 400 m shot offset. This feature is not observed in the isotropic case. Azimuthal variation in peak S-wave amplitude is also present but subtle.**

## 6 Discussion

Measurements of englacial fabric at Antarctic ice rises have the potential to help constrain recent ice sheet history. DAS VSP data and modelling results presented here highlight the potential of the method to improve our understanding of englacial fabric and therefore recent ice flow history.

Seismic velocities measured in the lower half of the ice column at SIR are high and indicate the likely presence of a strong vertical cluster fabric, consistent with models of fabric formation at ice rises. Observed relatively high amplitude S-waves from far offset shots are also consistent with the presence of englacial fabric. Uncertainties are however too high to establish the depth distribution or azimuthal symmetry of the anisotropy. A cluster fabric within the lower section of the ice column is consistent with ongoing ice divide flow and fabric formation mechanisms (Martin et al., 2009). The time required to form characteristic fabrics at ice divides is a function of the accumulation rate and ice thickness and as such varies widely. Stable flow over sufficient time is required to form the characteristic fabric. However, the low strain rate at ice rises also allows remnant fabric from a previous flow regime to persist (Brisbourne et al., 2019). The presence of a cluster fabric at depth is therefore not a conclusive indicator of long-term stable flow. Further data are required to investigate the presence of azimuthally variant fabric, for example, that may form under different flow regimes. Where more reliable measurements are available, the conversion of apparent to true velocity will require an estimate of fabric to account for velocity variation; the deviation of the raypath from the vertical will be a function of both fabric and source-receiver geometry. A correction will be

required to account for the fabric-induced difference in velocity between the raypath and the vertical. This is likely best posed as an inversion problem.

Our measured value of Q is low, indicating higher attenuation than previous field measurements in polar ice at similar temperatures, e.g. Q=400-670 at -14° to -25° C (Bentley, 1971) and Q=130-355 at -15° to -22° C (Peters et al., 2012). By invoking transformations to account for measurement frequency, impurity uptake from the mother solution and conversion from flexural vibration to acoustic seismic P-waves, Bentley and Kohnen (1976) demonstrated consistency between field measurements of seismic attenuation and the laboratory measurements of Kuroiwa (1964). However, the same laboratory measurements highlight the sensitivity of seismic attenuation to temperature, impurity concentration and signal frequency. An approximately exponential variation in attenuation occurs at temperatures close to the melting point (above -30° C) across all frequencies. Similarly, samples of ice from Antarctica and Greenland exhibit different behaviour to laboratory-generated samples in the temperature range -20° to -10° C. Laboratory measurements of internal friction at 4 to 9 Hz on the Mizuho ice core (Nakamura and Abe, 1978) also highlight sensitivity to temperature at the melting point and, in contrast to the measurements of Kuroiwa (1964), high degrees of internal friction are observed, equivalent to Q values as low as 10.

It is important to recognise that a number of mechanisms cause attenuation. Field-derived measurements provide an estimate of effective Q, the sum of intrinsic Q (e.g. material properties) and apparent Q (e.g. scattering). Decoupling the effects of intrinsic absorption from scattering (perhaps due to impurities or variations in ice crystal size or fabric) will be difficult without further information (e.g., derived from core). For example, Kuroiwa (1964) highlights the significance of grain boundaries and impurities in polycrystalline ice, which results in sensitivity to temperature close to the melting point. Similarly, using the flexural vibration method, Oguro et al. (1982) demonstrated that internal friction varies by up to a factor of five depending on the orientation with respect to the c-axis of a single ice crystal. Additional in situ measurements are therefore required to understand better these sensitivities. This requirement if highlighted by the microstructural analysis of the EPICA DML ice core (European Project for Ice Coring in Antarctica in Dronning Maud Land). There, grain size generally increases with depth but is strongly affected by the impurity content of ice from different climatic stages resulting in a step-change in grain size at depth (Weikusat et al., 2017), highlighting the shortcomings of attenuation profiles based on incomplete parameterisation.

Our measurements are unusual in that in situ measurements of temperature across the entire ice column are available: most seismic attenuation measurements acquired to date are not coincident with a measured profile of englacial temperature or physical properties of the ice. Furthermore, at SIR, the ice core will allow for measurements of impurity levels and potentially grain size, providing context to our field measurements. For example, mean NaCl levels in the SIR ice core are low (R. Mulvaney, pers. comm.), with concentrations below the lowest levels tested by Kuroiwa (1964), approximating pure ice for these purposes.

Unlike the DAS VSP observations of Booth et al. (2020), where direct P-wave spectra are consistent over a range of depths, energy loss with depth is apparent in our observations (Fig. 5c,d). Although the spectral ratio analysis method demonstrates that our attenuation measurement is robust, we cannot rule out the influence of the fibre optic cable hanging in the borehole on these measurements, potentially resulting in a form of signal amplitude reduction unrelated to ice properties. For example, is the uniformity of the coupling or behaviour of the cable influenced by the difference in hydrostatic head of drilling fluid above the sampled range? We therefore regard the signal loss as a robust observation but advocate further experiments to determine the cause of the high degree of attenuation.

Our ray tracing results indicate that englacial fabric at ice rises will likely lead to diagnostic features in DAS VSP data, such as azimuthal variation in traveltimes and amplitudes. We also observe shear wave splitting in the synthetic data, caused by the

girdle fabric. In addition to the fabric geometry, these modelling results are dependent on the velocity profile, seismic source geometry and the acquisition geometry, and will therefore act as a guide in the design of future experiments.

## 7 Recommendations

We combine our field experience, data analysis and modelling results to make a number of recommendations to assist in future
experiment design using DAS technology in polar borehole settings.

(1) With respect to the cable in the borehole, noise suppression and coupling to the ice control data quality. Unlike locations with high strain-rate or temperate ice, closure of boreholes at Antarctic ice rises will take a number of years. Therefore, in the time available, achieving a frozen-in cable to ensure both coupling to the ice and the suppression of harmonic vibrations is not always possible. In our data, below 350 m depth, we observe reflected P-waves and their surface-
multiples, implying that drilling fluid in the lower part of the borehole is providing good coupling to the ice. The harmonic noise is also suppressed over these depths. Therefore, where borehole closure will not occur naturally in a reasonable timeframe, the seismic experiment should be undertaken with drilling fluid filling the entire borehole (below the firn), i.e. prior to recovery of the drilling fluid. In addition, as variation in seismic amplitude is a key diagnostic for constraining fabric geometry, consistent coupling to the borehole is critical. Whether fluid provides this consistency remains untested.
An alternative solution may be backfilling of the borehole with water to accelerate freeze-in.

(2) Although data acquired with the hammer and plate seismic source demonstrate the feasibility of the DAS method, and have been demonstrated as viable where the fibre optic cable is well-coupled (Booth et al., 2020), a more energetic source (e.g., explosives) would increase the signal to noise ratio and improve the analysis capability beyond that demonstrated here.

(3) Using a geophone and data logger located adjacent to the hammer plate proved to be adequate for estimating source time in this experiment. However, improved timing accuracy is possible with either direct trigger of the DAS recording unit or higher sampling rate of the geophone adjacent to the source. Alternatively, if using an explosive source, we recommend a GPS-synchronised trigger in parallel with GPS timing on the interrogator.

(4) A ruggedised interrogator unit would reduce complexity of the acquisition process (e.g., tolerance to temperature;
ruggedized user interface; tolerance to unplanned power outage with automatic restart). An interrogator with low power consumption, or a substantial solar power system, would enable the use of battery power and thus remove the noise source of the generator. In this case, if the interrogator unit requires mains power (110 or 240 V AC) then an inverter is required, the use of which needs to be carefully managed in polar environments to protect both the user and the instrument.

(5) This experiment utilised a multi-mode fibre cable deployed specifically for DTS measurements. However, it is now
standard to deploy cables with both multi-mode and single-mode fibres and this should be utilised for all future deployments. Optical fibre with a helical structure may extend measurements beyond one dimensional longitudinal strain (Kuvshinov, 2015).

(6) Modelling of the seismic response to likely fabric configurations will optimise field acquisition. Although amplitude variation is a key diagnostic in the scenarios tested here, strong girdle fabrics will also result in measureable traveltime
variation.

(7) Modelling results for likely ice rise fabric indicate that S-wave traveltime and amplitude variations are key diagnostics with the seismic method. Reliance on P-wave traveltime and amplitude variation will limit the capacity of the observations to constrain fabric. As such, a reliable S-wave seismic source is essential (e.g., Lutz et al., 2020). To what degree an explosive source will be effective remains to be tested.

(8) Additional data, such as polarimetric radar measurements or a three-component geophone surface array may improve uniqueness of any interpretation. The deployment of additional fibre optic cable along the surface will complement

downhole observations at little logistical overhead. However, measurements using surface-deployed cable will be limited to horizontal longitudinal strain rate and further modelling will be required to explore the benefit of this additional information in such scenarios. With a single fibre optic cable for both downhole and surface measurements, only a single interrogator unit is necessary.

## 8 Conclusions

We have demonstrated the feasibility and potential of DAS measurements in Antarctic settings using existing infrastructure. This presents opportunities for relatively low-cost and logistically light experiments, both adding value to existing studies and opening opportunities for continent-wide experiments at relatively low risk. We have presented results from a walkaway VSP experiment acquired at SIR. Estimates of seismic velocity ($3984 \pm 218$ m s$^{-1}$) in the lower section of the ice column are consistent with a cluster fabric. Relatively high S-wave amplitudes from far-offset shots are also consistent with the presence of fabric. Estimates of Q in the same interval ($75 \pm 12$) are lower than previous estimates of attenuation in ice. Our data lack the signal strength required to allow a more detailed interpretation of the structure at SIR. However, observations of direct and reflected seismic waves from a low-energy hammer source at a range of offsets and azimuths underscore the potential of the DAS method to investigate englacial and basal structure in similar settings. Modelling of the effect of likely ice fabric geometries on P- and S-wave traveltimes and amplitudes demonstrates the potential for this method to identify and discriminate ice fabric at ice rises. The results of this pilot study present an opportunity to improve the design of future experiments and we make a number of recommendations related to both acquisition design, instrumentation and field methodology.

## Data availability

Data are available at the Polar Data Centre at https://www.bas.ac.uk/data/uk-pdc/ for public access (Brisbourne and Kendall, 2021).

## Author contribution

AB and MK designed and led the experiment with support from SK and AS. MK conducted data acquisition with support from AB and SK. AB undertook data analysis with assistance from MK, TH and SK. AB wrote the manuscript with contributions from all others.

## Competing interests

The authors declare no competing interests.

## Acknowledgements

We are grateful to Anne Flink, BAS Operations and Rothera Station personnel who supported this experiment. Athena Chalari and Andy Clarke at Silixa Ltd loaned equipment, assisted with planning and provided technical support. Fieldwork was undertaken as part of the BEAMISH Project (NERC AFI award numbers NE/G014159/1). JMK was supported by additional funding from NERC award CASS-166. JMK and TSH were partially funded by the DigiMon project (project no. 299622), which is part of Accelerating CCS Technologies (ACT) programme. The SIR borehole and fibre optic cable are part of the University of Cambridge WACSWAIN Project (EU Horizon 2020 agreement No. 742224). We thank the Computational

Infrastructure for Geodynamics (http://geodynamics.org) which is funded by the National Science Foundation under awards EAR-0949446 and EAR-1550901.

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
