# Peer review of "Downhole distributed acoustic seismic profiling at Skytrain Ice Rise, West Antarctica"

_The Cryosphere, 2021_

## Referee Comment (RC2)

Review: Brisbourne et al., Downhole distributed acoustic seismic profiling at Skytrain Ice Rise, West Antarctica.
March 2021
* * *
Brisbourne et al present results from the first Antarctic Distributed Acoustic Sensing (DAS) experiment. The manuscript is a useful contribution. It focuses on a novel method with the potential to make a significant scientific contribution. The preliminary results presented provide some constraints on the compressional wave velocity of the lower half of the ice column, and an estimate of seismic quality factor. The manuscript's main intent is to act as guide for future experiments and it provides an extensive list of recommendations. The manuscript is generally well written and presented but there are a few areas that would benefit from improvement. My main comments pertain to communicating the justification for this type of experiment, discussion/comparison with other borehole seismic methods, and consistency in the data presented.

**1. Main comments**

**Introduction**

The introduction could be improved by better justifying this style of experiment. First, the connection between velocity, amplitude, and crystal orientation fabric (COF) needs to be outlined. Readers will be confused by the use of a seismic method where ice core physical properties are available. To address this, the introduction should emphasise where DAS is possible but direct measurements are not. The introduction should also outline what DAS brings that other seismic methods don't. This will require other borehole seismic methods (clamped borehole seismometers, direct measurements of recovered core, acoustic borehole logging) to be summarised. Highlighting the suitability of DAS deployments in irregular hot water drilled holes where other seismic methods are not possible is a real selling point for this method. The other justification often used is to inform surface observations and improve surface based methods. Does DAS provide any advantages here? The description of fabric evolution is also brief. Presenting the typical ice divide COF progression would be helpful here as would a mention of how impurities, temperature, and strain, influence fabric evolution. Addressing these points should make the manuscript more accessible. At present the first two paragraphs of the introduction are not so relevant for the rest of the manuscript, although a focus on COF could make them so.

**Data and results presented**

Data are presented from a range of offsets. It would be helpful if a consistent set of offsets were used. 0, 200, 400, 600 m would makes sense. As it is, in Fig. 2 we see the bandpassed checkshot with and without FK&Decon, then a zoom of the 100 m shot, then the 500 m offset shot. In

Fig 3. we see the 150 m shot and synthetics. Then in Figure 4 we see results from the 0, 50, and 100 m shots and in Figure 6 we see estimates for 200 m and 400 m offsets. Presenting the same offsets make it easier to follow along and give the reader more confidence.

The diamond shaped noise source is nicely explained.

As these data are new to most of us, it would be good to see the waveform of the arrival. In my experience the devil is in the picking. It would be instructive to see waveform wiggles overlain with picks. After conversion to velocity would be the most useful.

Is it possible to present the results in Figure 6 in a similar way to the field data displayed in Figures 2, and 3? If so it would make interpretation by the readers much easier.

**2. Minor points**

L46 'gravitaionally driven' is too general. Be explicit about what's not going on and why that's useful.

L47 'preserve recent' and the not so recent. COF evolution depends on the existing state. Unravelling the strain history is not as straightforward as this statement suggests.

L54 'as with all surface geophysics' is a sweeping statement. Again be explicit.

L138 and Fig2 b) reverse-moveout coherrent noise has made it through the FK filter implying the not just positive dips preserved or maybe filter tapers.

L151 'snow compacting' – snow compacting and metamorphosing.

L198-199 First 2 sentences of this para belong in the introduction.

L210-212 What is the impact of the assumption of straight ray-paths. It would be good to assure the reader this is insignificant.

L238–244. If I follow this correctly each trace is replaced by a stack from a 10 m bin after the removal of traces that fail to cross correlate at $> 0.95$. With this procedure if the central trace is the outlier trace it will remain dominant. Also, this stacking will lower the frequency content. Will this change the result? Regardless of this the reader should know what percentage of traces were removed by the editing procedure.

L264-265 '...and seismic methods provide...' citation needed.

L282 'Skytrain' – SIR (for consistency).

L287 refer to Fig 6 c).

L287–290. Please elaborate on this. If possible, seeing these results in the same gather form as Figs 2  3 would be very helpful.

L293 '..very small' How small? Possible to pick in real data?

L376-377 'multimode' – multi-mode (for consistency) also introduce/define single-mode and multi-mode and elaborate on benefits.

L384-385 'As variation....is therefore critical' Combine this with reccomendation 1).

**3. Figures**

See comments above regarding presenting similar offset shots and results.

Figure 1. Coordinates required on either b) or c), preferably both.

---

## Author Comment (AC1)

**Response to reviewer comments: "Downhole distributed acoustic seismic profiling at Skytrain Ice Rise, West Antarctica" by Alex M. Brisbourne et al., The Cryosphere Discuss., TC-2021-1"**

We thank the editor and reviewers for their constructive and valuable comments on the manuscript. We feel that our revised manuscript has been greatly improved and made more accessible to a broader community. We directly address the comments below with our responses highlighted in red.

Both reviewers commented on the presentation style of the synthetics. Our response to this has resulted in the addition of Figure 7, and also forced us to re-analyse our synthetic results. This led us to examine the depth distribution of traveltimes in more detail and identify greater azimuthal variation than previously recognised. This process also highlighted an incorrectly parameterised S-wave velocity gradient in the model. We have therefore reproduced Figure 6 with synthetics using the corrected velocity profile. The trends in results do not change but the magnitude of the amplitude anomalies is now greater and we discuss this further in the body of the text. Although the results have not changed we can now conclude that both amplitudes and traveltime anomalies may both be diagnostic measurements of ice divide fabric.

**Reviewer #1**

**General Comments**
This paper presents results and lessons learned from a distributed acoustic sensing (DAS) experiment at Skytrain Ice Rise in the Weddell Sea Sector of West Antarctica. Although, in the upper part of the borehole (~0-350 m) noise from the cable vibrating in the hole is too large to accurately recovery seismic interval velocities. At greater depths (>350 m), when the cable is suspended in drilling fluid, the P wave seismic interval velocity and attenuation are measured. In this study, the data acquired are too noisy to delineate any englacial fabric at Skytrain Ice Rise. The authors summarise with a list of recommendations for future use.

This paper is well written, clear and concise. Although, the data quality was not good enough to delineate any englacial fabric, the authors explained what they could observe and calculate from the data evidently. However, the authors only calculate the seismic Pwave velocity, yet in Figure 2 and 3 they point out the down going S-wave. The authors should explain why they do not also calculate the S-wave velocity, which they show later on in the synthetic modelling to be affected the most by changes in englacial fabric.

The authors summary, which includes a list of recommendations and lessons learned, is an important component of the paper. It details a list of recommendations for future use of DAS on ice which is important to share with the Cryosphere community. However, some of the figures and explanations could be made clearer to make this paper more reader friendly. Some specific comments are detailed below and I hope they are useful.

**Specific comments**
*Figure 1 and Line 97-102 :* A diagram (photo or schematic) might be nice to explain the DAS set up more clearly including the description of the bend/splice in the cable (line 101-102). Since this is a pilot study and only been applied twice down boreholes through ice, the cryosphere community are not familiar with this type of acquisition and therefore it would be nice to see potentially a photo or schematic diagram of the set up. This could be figure 1.d.
We have added a photo of the DAS interrogator setup in the tent and a schematic of the system layout in Figure 1. We have added additional text in the introduction to describe the DAS system, including the bend in the fibre.

*Line 110 – 112 :* "The interrogator unit recorded in continuous mode with shot gathers subsequently extracted using times derived from impulsive arrivals on a continuous 1000Hz sampled geophone recording made adjacent to the hammer plate." Can you clarify this sentence in a bit more detail, in particular the second part of the sentence I am struggling to follow it. How are the times derived from impulsive arrivals? The geophone recording you mention is the continuous geophone string of the DAS cable?

This has been clarified in the text: "Hammer impact times were derived from a continuous 1000 Hz record of a GPS-synchronized 3-component surface geophone deployed adjacent to the hammer plate."

*Line 119 – 122 :* Sentence starting with "From zero to 150 ms, ...." Can you add an interpretation of where this is on the image Fig 2a. It is not obvious what you are referring too. Am I correct in understanding the strong red-blue-red horizontal signal (0-50 ms) in Figure 2b is caused by wind and generator noise affecting the interrogator unit? If so this should also be clarified in this sentence and pointed out in Fig 2b.0

You are correct. We have added annotation to Fig. 2.

*Figure 2:* It would be nice to see a raw frequency spectrum of the data shown in Fig 2.a.

We now include example waveforms of five traces below the P-wave arrival discontinuity. We do not present spectra as Figure 2 presents bandpass filtered data and we use the Q measurement section to display the spectra.

I see you say it in the figure citation text but visually it would be easier to interpret c) if you had a thin box around your zoomed in area in a). I understand you might not have wanted to clutter image a) but it took me a while to put c) into context as it is plotted the same size as a) and b), which is a bit confusing with no window plotted.

We no longer plot this section as a zoomed version as this appeared to be confusing. We therefore maintain consistent scaling on all plots.

a), b) and c) are the zero offset but d) is when the source is at 500 m offset. Again, it might be clearer to add labels on the top of a) b) and c) "0 m offset" and label d) "500 m offset" in the image for clarity.

Done. Also, in response to Reviewer #2 we now use 0/200/400 m offsets for consistency.

Add an interpretation of the red-blue-red strong horizontal signal at the top of figure 2.b around 0-50 ms.

Done

In figure 2.a if that is a P-wave multiple does it arrive at the times you would expect for a multiple?

Yes it is. We have removed the question mark from the figure label.

What causes the dimming of the background noise (the white section) between 0 and ~150 m depth?

Traces are normalised to the maximum amplitude at each depth. Due to the relatively high amplitude at shallow depths (0-150 m, 0-60 ms) this results in an apparent dimming of the signal to 150 m depth. We persist with this scaling as it aids visualisation and have added a note to the figure caption to state this.

What causes the discontinuity in the P- wave in figure 2a, which is in-line with the end of the dimmed white section?

This is a result of the dominance of the high amplitude signal which immediately follows the P-wave arrival and highlights the issues with these data. We have added a sentence to reflect this.

*Line 154*: "The low temperature… " what is low? can you provide a temperature range? Also, if you have borehole information, e.g. a temperature and strain profile, it might be worth adding these profiles to one of the figures.
SIR temperature profile data are published in Mulvaney (2020) and we present only the relevant temperatures here. We clarify "low" as -26C at 10 m depth.

*Line 156:* (Mulvaney, 2020) reference. This paper is in review in AoG and therefore I could not access it for reviewing statements referencing it.
Currently in print.

*Figure 3:* This is very clear and great observation/simulation to explain the source of noise.
Thank you

*Line 170:* can you mark this discontinuity in Fig 3a.
We have added a label to 3a and 3c

*Line 192 and 194:* "… open parts of the borehole…." And " …. deeper parts of the borehole." Add depth ranges to remind the reader. (0-350 m ?) and (350 – 595 m ?)
Done

*Section 4.1 Seismic P-wave velocity:* In Figure 2 and 3 you point out the down going Swave. Yet in this section you only measure the P-wave velocity. Is your downgoing Swave too weak and below the background noise level to measure it's velocity? You should state why you don't measure the S-wave velocity somewhere in the paper.
Correct. We have added the text "The signal to noise level is too low to derive reliable S-wave velocities."

*Line 207:* "Noise in the upper half…." Is this the harmonic noise caused by the vibrating cable? might be good to clarify that.
Correct. We have clarified this in the text.

*Line 208:* Why is the noise so consistent? Is it because it is repeatable and constantly the same, from the vibrations of a repeatable hammer and plate source?
Our wording here has caused confusion here and we have clarified the text. We did not mean to indicate that the harmonic noise is consistent, only that the anomalies in the velocities related to the discontinuity in the P-wave arrival are consistent. The discontinuity in the P-wave at 300 m depth is consistent as we already noted in Section 3.1.

*Line 209:* It might be useful to state what exact velocities you mean here. I am guessing you are talking about the velocities > 5000 m/s.
Our wording of how we selected this window was confusing. We now state that we select this window to exclude the harmonic noise and discontinuity in the P-wave arrival.

*Line 222:* "Although uncertainties are large…." Reference table 1 here and add sentence about how this compares with uncertainties expected from other seismic methods for deriving Vp.
We have added a reference to Table 1 and highlighted that they are comparable to uncertainties in previous studies in similar environments.

*Line 223:* "… most likely a vertical cluster fabric" could you elaborate a little bit more. Maybe add an extra sentence explain what this is and why you have come to that observation.

We have added text to clarify this statement and included further information on fabric types in the introduction.

*Equation 1:* Need to define the variables clearly in the text.

Apologies, this was an oversight on our part. Done.

*Line 238:* Averaging vertically over 10 m… where does this "10m" come from? Are there any references for this? I thought DAS had very good vertical resolution and therefore the averaging window was much smaller than 10 m (but I am not a DAS expert!)?

10 m is the gauge length of the Silixa iDAS system deployed here. We have added text to the introduction to describe gauge length and more details on the DAS method.

*Figure 5 and Line 241:* Here you mention upper and lower traces, I don't see any mention in the text of where upper and lower are? What is the window "upper" and "lower" traces are taken from on the shot gather?

We describe the depth interval over which the spectral ration method is applied (380 to 540 m), consistent with the velocity analysis and highlighted in Figure 4. We now also add these depths explicitly in the caption to Figure 5 to avoid confusion.

*Line 246-247:* It might be useful to have mentioned this earlier on when taking about "low temperatures".

Apologies for this. We have added the 10 m depth temperature at the earlier stage.

*Line 271-272*: Do you input Q to this modelling?

We use a standard value of Q in the calculation. However, scaling of the harmonic noise dominates the summed trace. We tested values of Q in SpecFEM and it is apparent that within a reasonable range the attenuation values assigned in SpecFEM have minimal influence at the scale of plot in Fig. 3a. As this section is simply to demonstrate the possibility that the noise is harmonic motion and not create a useable synthetic we do not discuss this point in the manuscript.

Line 281: "a more energetic seismic source" like explosives or a low frequency source like a vibroseis for the S-waves? maybe add an example of what source you would recommend.

We refer to the source used by Lutz et al. (2020) which also utilises a hammer.

*Line 296:* How would you, therefore, maximize your S-wave signal to be able to observe this on real data? Do you need a different/lower frequency source? or do you just need to get lucky and have really low noise?

The S-waves need to be reliably generated at source and we have added a reference to a suitable device under the recommendations.

Figure 6: S1 amplitude (400 m offset) are difficult to compare directly as the amplitude scale is different for them all. In a) it is 0.3-0.5 in b) it is 0.1 - 0.9 and c) 0.1-0.8 I would suggest having a different colourbar for traveltime plots and amplitude plots.

We accept that as this is a multi-dimensional and multi-parameter modelling exercise there are a number of ways to display these results and the best method may differ to the one presented here depending on the mode of analysis. The significance of each plot in our presentation is that they represent a single ice rise fabric structure. The important comparison is therefore within a single fabric structure rather than between different structures. For example, to distinguish an isotropic structure from a cluster/girdle one would look for azimuthal variations in SV amplitude. It is the

variation with azimuth which is important, the relative strength between the two. Likewise, for the cluster only fabric it is the depth of the amplitude peak at a given azimuth and offset which is the key diagnostic, not the relative amplitude between offsets. Amplitudes calculated with the ray tracing are all relative and depend on an assumed isotropic source function and perfect and identical receiver coupling. We therefore persist with our scaling method to highlight the desired features and accept that improvements may be made if different perspectives are required, such as that suggested by Reviewer #2 and presented in Figure 7.

*Section 7. Recommendations:* Great section and very useful for future research using DAS.
Thank you

*Recommendation 3:* this would be easier to understand if there was a schematic diagram of the set up in Figure 1 (as mentioned above).
We have added a schematic of the field set up to Figure 1.

*Recommendation 5:* Do you deploy these bundles down the borehole all at once? So you have multiple cables down one hole? ... or have I interpreted that incorrectly?
This is both fibre types within a single cable. We have clarified the text to reflect this more clearly.

Also, "… and this should 'be' utilised …". I think you are missing a "be" in the last part of the sentence.
Done

*Recommendation 7*: "As such, a reliable S-wave seismic source is essential." For example …. ?
We refer to Lutz et al. (2020)

---

## Author Comment (AC2)

**Response to reviewer comments: "Downhole distributed acoustic seismic profiling at Skytrain Ice Rise, West Antarctica" by Alex M. Brisbourne et al., The Cryosphere Discuss., TC-2021-1"**

We thank the editor and reviewers for their constructive and valuable comments on the manuscript. We address the comments below with our responses highlighted in red.

Both reviewers commented on the presentation style of the synthetics. Our response to this has resulted in the addition of Figure 7, and also forced us to re-analyse our synthetic results. This led us to examine the depth distribution of traveltimes in more detail and identify greater azimuthal variation than previously recognised. This process also highlighted an incorrectly parameterised S-wave velocity gradient in the model. We have therefore reproduced Figure 6 with synthetics using the corrected velocity profile. The trends in results do not change but the magnitude of the amplitude anomalies is now greater and we discuss this further in the body of the text. Although the results have not changed we can now conclude that both amplitudes and traveltime anomalies may both be diagnostic measurements of ice divide fabric.

**Reviewer #2 - Hugh Horgan**
Brisbourne et al present results from the first Antarctic Distributed Acoustic Sensing (DAS) experiment. The manuscript is a useful contribution. It focuses on a novel method with the potential to make a significant scientific contribution. The preliminary results presented provide some constraints on the compressional wave velocity of the lower half of the ice column, and an estimate of seismic quality factor. The manuscript's main intent is to act as guide for future experiments and it provides an extensive list of recommendations. The manuscript is generally well written and presented but there are a few areas that would benefit from improvement. My main comments pertain to communicating the justification for this type of experiment, discussion/comparison with other borehole seismic methods, and consistency in the data presented.

1. Main comments
Introduction
The introduction could be improved by better justifying this style of experiment. First, the connection between velocity, amplitude, and crystal orientation fabric (COF) needs to be outlined. Readers will be confused by the use of a seismic method where ice core physical properties are available. To address this, the introduction should emphasise where DAS is possible but direct measurements are not. The introduction should also outline what DAS brings that other seismic methods don't. This will require other borehole seismic methods (clamped borehole seismometers, direct measurements of recovered core, acoustic borehole logging) to be summarised. Highlighting the suitability of DAS deployments in irregular hot water drilled holes where other seismic methods are not possible is a real selling point for this method. The other justication often used is to inform surface observations and improve surface based methods. Does DAS provide any advantages here? The description of fabric evolution is also brief. Presenting the typical ice divide COF progression would be helpful here as would a mention of how impurities, temperature, and strain, influence fabric evolution. Addressing these points should make the manuscript more accessible. At present the first two paragraphs of the introduction are not so relevant for the rest of the manuscript, although a focus on COF could make them so.
We have reworked and extended the entire introduction to address the shortcomings highlighted by the reviewer.

Data and results presented
Data are presented from a range of offsets. It would be helpful if a consistent set of offsets were used. 0, 200, 400, 600 m would makes sense. As it is, in Fig. 2 we see the bandpassed checkshot with and without FK&Decon, then a zoom of the 100 m shot, then the 500 m offset shot. In Fig 3. we see the 150 m shot and synthetics. Then in Figure 4 we see results from the 0, 50, and 100 m shots and in

Figure 6 we see estimates for 200 m and 400 m offsets. Presenting the same offsets make it easier to follow along and give the reader more confidence.

This is a good point. We now present 0/200/400m in Figure 2. We have re-run the synthetics in Figure 3 at 200 m offset for consistency. This is also then consistent with Figure 6 which presents 200/400 m. By its nature, Figure 4 requires the nearest offsets to obtain reliable velocities so we retain 0/50/100 m offsets here.

The diamond shaped noise source is nicely explained.

Thanks

As these data are new to most of us, it would be good to see the waveform of the arrival. In my experience the devil is in the picking. It would be instructive to see waveform wiggles overlain with picks. After conversion to velocity would be the most useful.

We have included sample example waveforms Figure 2. As velocities are calculated using relative arrival times we do not rely on first breaks for arrival times but use peak arrivals. We have added this to the body of the text.

Is it possible to present the results in Figure 6 in a similar way to the field data displayed in Figures 2, and 3? If so it would make interpretation by the readers much easier.

We now present synthetic VSP gathers of the isotropic and girdle/cluster fabric in an additional figure (Figure 7). This process has led us to investigate the synthetics in more detail, with some interesting conclusions, which we now discuss in more detail.

2. Minor points

L46 `gravitaionally driven' is too general. Be explicit about what's not going on and why that's useful.
L47 `preserve recent' and the not so recent. COF evolution depends on the existing state. Unravelling the strain history is not as straightforward as this statement suggests.

We have expanded the introduction to include the description and referencing of ice flow at ice rises.

L54 `as with all surface geophysics' is a sweeping statement. Again be explicit.

We have reworked and extended the entire introduction to address the shortcomings highlighted by the reviewer.

L138 and Fig2 b) reverse-moveout coherrent noise has made it through the FK filter implying the not just positive dips preserved or maybe filter tapers.

This is the most likely explanation. We have added a sentence to the figure caption.

L151 `snow compacting' - snow compacting and metamorphosing.

Done.

L198-199 First 2 sentences of this para belong in the introduction.

We have reworked and extended the entire introduction to address the shortcomings highlighted by the reviewer.

L210-212 What is the impact of the assumption of straight ray-paths. It would be good to assure the reader this is insignificant.

Good question. We have now ray traced with the isotropic velocity model to determine the angle at the fibre for the calculation of true velocity from apparent. Calculation of the true angles is an

inversion problem as it requires a velocity model fit specifically to these data. Unfortunately, our data are insufficient to determine a fully anisotropic velocity model.

L238-244. If I follow this correctly each trace is replaced by a stack from a 10 m bin after the removal of traces that fail to cross correlate at > 0.95. With this procedure if the central trace is the outlier trace it will remain dominant. Also, this stacking will lower the frequency content. Will this change the result? Regardless of this the reader should know what percentage of traces were removed by the editing procedure.

If the central trace is an outlier it will result in most if not all of the adjacent traces being discarded due to low coherency, and therefore not produce a result. We now report the number of traces removed by the 0.95 threshold (45%). The DAS system deployed here utilises a gauge length of 10 m so the selection of a 10 m window is consistent with this and will not change the frequency content further.

L264-265 `...and seismic methods provide...' citation needed.
Done. Brisbourne et al. (2019)

L282 `Skytrain' - SIR (for consistency).
Done

L287 refer to Fig 6 c).
Done

L287-290. Please elaborate on this. If possible, seeing these results in the same gather form as Figs 2 3 would be very helpful.
We have created Figure 7 which presents synthetic VSP gathers for the isotropic and divide fabric cases to highlight some of the features expected. We discuss these details further in the text.

L293 `..very small' How small? Possible to pick in real data?
Good question. We have added "However, in our model the effect is small. For example, the azimuthal variation in P-wave traveltimes is less than 4 ms with either a 200 or 400 m source offset."

L376-377 `multimode' - multi-mode (for consistency) also introduce/define single-mode and multi-mode and elaborate on benefits.
We have added more detail to the introduction describing the DAS method, including fibre types.

L384-385 `As variation....is therefore critical' Combine this with recommendation 1).
Done

3. Figures

See comments above regarding presenting similar offset shots and results.
See responses above.

Figure 1. Coordinates required on either b) or c), preferably both.
We have added the borehole location to the figure caption to avoid clutter on the figures.

---

## Author Response (AR3)

**Author response to Editor Decision: Publish subject to technical corrections (24 Jun 2021)**

**Alex Brisbourne, 28ᵗʰ June 2021**

We are grateful to the editor and anonymous reviewer for the additional time they have spent reviewing the updated version of our manuscript. We have addressed the "technical corrections" in line with all recommendations, as outlined below.

p. 5, line 74: Should say "... when an ice core has not been recovered..."?

Done.

Figure 7: c) top plot: should this not be azimuth 0 not 90 ?

Well spotted! Corrected.

Figure 7: I would label P-wave and S-wave in one of the boxes, maybe a) top left plot (isotropic) just to help the reader get their eye in quickly. I would also label the shear wave splitting in c) middle plot again to guide the reader. This is a nice clear example of the shear wave splitting.

Good idea. Done.

Figure 6: P wave traveltime is this in seconds? Maybe worth adding this.

Yes, done. In addition, we have clarified at L328 that amplitudes from the ATRAK software are displacement.